# Role of Peptides in Diagnostics

**DOI:** 10.3390/ijms22168828

**Published:** 2021-08-17

**Authors:** Shashank Pandey, Gaurav Malviya, Magdalena Chottova Dvorakova

**Affiliations:** 1Department of Pharmacology and Toxicology, Faculty of Medicine in Pilsen, Charles University, 32300 Pilsen, Czech Republic; 2Cancer Research UK Beatson Institute, Garscube Estate, Switchback Road, Glasgow G611BD, UK; g.malviya@beatson.gla.ac.uk; 3Department of Physiology, Faculty of Medicine in Pilsen, Charles University, 32300 Pilsen, Czech Republic; magdalena.dvorakova@lfp.cuni.cz; 4Biomedical Center, Faculty of Medicine in Pilsen, Charles University, 32300 Pilsen, Czech Republic

**Keywords:** peptides, diagnostic, ELISA, microarray, PET, SPECT, imaging diagnostic, non-imaging diagnostic

## Abstract

The specificity of a diagnostic assay depends upon the purity of the biomolecules used as a probe. To get specific and accurate information of a disease, the use of synthetic peptides in diagnostics have increased in the last few decades, because of their high purity profile and ability to get modified chemically. The discovered peptide probes are used either in imaging diagnostics or in non-imaging diagnostics. In non-imaging diagnostics, techniques such as Enzyme-Linked Immunosorbent Assay (ELISA), lateral flow devices (i.e., point-of-care testing), or microarray or LC-MS/MS are used for direct analysis of biofluids. Among all, peptide-based ELISA is considered to be the most preferred technology platform. Similarly, peptides can also be used as probes for imaging techniques, such as single-photon emission computed tomography (SPECT) and positron emission tomography (PET). The role of radiolabeled peptides, such as somatostatin receptors, interleukin 2 receptor, prostate specific membrane antigen, αβ3 integrin receptor, gastrin-releasing peptide, chemokine receptor 4, and urokinase-type plasminogen receptor, are well established tools for targeted molecular imaging ortumor receptor imaging. Low molecular weight peptides allow a rapid clearance from the blood and result in favorable target-to-non-target ratios. It also displays a good tissue penetration and non-immunogenicity. The only drawback of using peptides is their potential low metabolic stability. In this review article, we have discussed and evaluated the role of peptides in imaging and non-imaging diagnostics. The most popular non-imaging and imaging diagnostic platforms are discussed, categorized, and ranked, as per their scientific contribution on PUBMED. Moreover, the applicability of peptide-based diagnostics in deadly diseases, mainly COVID-19 and cancer, is also discussed in detail.

## 1. Introduction

The development of accurate diagnostic methods is an urgent need in today’s world. Due to the upsurge of various deadly diseases, rare diseases, and cancer, it is crucial to improve the diagnostic aspects, which will help the clinician to predict and examine therapeutic responses across a wide spectrum of diseases.

In the past few decades, immunodiagnostics has been an essential tool for clinical management and prognosis of a disease. To discover novel biomarkers, it is obligatory to understand the effect of a disease on the physiology of organisms, as well as their impact on genomic and proteomic patterns. In some scenarios, the development of new diagnostics is limited because of already known and well-characterized biomarkers. On the contrary, mapping of protein antigen for selection of linear epitopes by peptide scanning is a widely used technique [1,2]. Moreover, rapid development in peptide microarray technology has further advanced the screening platform for serological screenings [3]. To select, identify, and design immunodominant linear or continuous epitopes by scanning all the predicted protein sequences using bioinformatics approaches is easy for an effective, rapid, and inexpensive way to validate the diagnostic markers.

The first systematic method for identifying T- and B-cell epitopes was the PEPSCAN method [4,5,6,7,8,9]. However, the majority of diagnostic assays developed are based on antigen–antibody reactions, and diagnostic assays are limited to antigenic sites of antibodies; but, T-cell epitopes can also be defined equally well using similar methods [10]. 

Moreover, there are a number of methods to determine linear B-cell epitopes. Generally, epitopes are of two types: (1) continuous epitopes (i.e., epitopes are derived from the epitope-mapping experiments of antigenic protein sequences); and (2) discontinuous epitopes (i.e., epitopes are identified by screening of complex peptide libraries [5]. Common methods used for selection of linear epitopes are (1) prediction (by using algorithms); (2) epitope recognition; (3) mutation in antigenic sequence or ‘escape mutants’ of viruses; and (4) PEPSCAN, by which overlapping peptides are tested for their ability to bind the antibody. The most systematic and reliable method for identifying linear antigenic peptides among all four methods described earlier is PEPSCAN [7]. However, a linear epitope can also be selected by using Methods 1 to 3.

On the contrary, structural epitopes are also screened by using combinatorial peptide libraries, which can be comprised of myriad peptide variants of either chemical or biological origin. Phage display is a powerful strategy that includes three steps to create a peptide library to screen functional peptides and proteins for specific biological functions. To create a peptide library, random DNA sequences are inserted into genes encoding protein 3 (cPIII) or protein 8 (cPVIII) of the filamentous phages. The library is first screened negatively against non-specific ligands and then positively against the desired target in vitro and in vivo. The identified peptides will be chemically synthesized and validated. The detailed principles and practices have been excellently reviewed by Smith and Perenko [11].

Recently, Songprakhon and co-authors identified 11 different sequences of 12-mer peptides binding to dengue virus nonstructural protein 1 by using a phage-displayed peptide library [12].

In addition to phage display, an alternative strategy is combinatorial peptide libraries that generate functional peptides. Huge peptide libraries can be established by peptide synthesis techniques for screening of unique ligands. Moreover, combinatorial peptide libraries are advantageous because non-peptidic moieties, such as beta-amino acids, un-natural amino acid analog, and modified peptide residues (phosphorylated or glycosylated), can be incorporated into the peptide sequences. The detailed principles and practices have been excellently reviewed by Bozovičar and Bratkovič in 2019 [13], where the current trends of peptides in imaging and non-imagining diagnostics are described.

## 2. Role of Peptides in Diagnostics

To understand the role of peptide in diagnostics, we have thoroughly investigated the published literature of last decade (*w.e.f.* 1 January 2011 to 31 December 2020) on the PUBMED MEDLINE database using specific keywords such as “Diagnostic” along with two filters “protein” and “peptide”. Although, the data acquired from these searches were based on algorithms and the results were dependent on the mapping of the articles/reviews/clinical trials and its match with specific words. However, many interesting facts were found during the scrutiny of the published data. In our search of the published articles in last decade (2011–2020) versus the total data published (1997–2022), we did not observe any big differences in the trend of using peptides versus proteins in diagnostics. Uses of peptides are always 2.5 times lower than proteins as per the published literature on PUBMED (Figure 1A1). We have also observed that use of peptides in diagnostics are constant and has been showing linear growth as per data published in 1 years, 5 years, and 10 years on PUBMED (Figure 1B1). The published literature on PUBMED for the last 1 year, 5 years, and 10 years has shown 18,963, 149,130, and 332,657 articles, respectively (Figure 1B2).

To further understand the role of peptides in diagnostics and get a clear picture of the usage of peptides in diagnostics, we had critically analyzed our extracted data for the last decade (*w.e.f.* 1 January 2011 to 31 December 2020) on the PUBMED MEDLINE database using specific keywords, such as “Peptide” with three additional filters such as “Drug” or “Vaccine” or “Diagnostic”. Data acquired from these searches were based on algorithms and the results were dependent on the mapping of the articles/reviews/clinical trials related to the keywords as mentioned above. It may contain some redundant data, due to the limitation of the analysis. However, some very interesting facts were found during the analysis, such as the total number of published scientific literature on PUBMED using the keyword “Peptide” along with additional filters such as “Drug” or “Vaccine” or “Diagnostic”, which was 440,613, and 25,399, and 347,534, respectively. The data confirm that the use of peptides in drug was 1.26 times higher than peptides in diagnostics. However, peptides in diagnostics were 13.7 times higher than peptides in vaccines (Figure 2).

## 3. Non-Imaging Diagnostics

Accurate and rapid detection of any diseases in humans has been a continuous challenge to diagnostic and epidemiological research. Efficient diagnosis is a crucial step, which helps in making an effective disease management strategy. A multitude of approaches have been attempted to identify pathogenic viruses and bacteria by using antigenic synthetic peptides in serological and molecular assays. Detection assays, which are based on peptides, have become increasingly substantial and indispensable for its advantages of using short synthetic peptides over conventional methods using recombinant proteins. Synthetic short peptide ligands with a length of more than eight amino acids have various advantages in the detection of specific antibodies [14].

To understand the role of peptides in non-imaging diagnostics, we have analyzed the published literature on PUBMED for last 5 decades (1 January 1970 to 31 December 2020). Non-imaging techniques such as ELISA, microarray, biosensors, microfluidics, and multiple Reaction monitoring were compared on PUBMED using keyword “peptide diagnostic”. As per data published on PUBMED, we observed ELISA ranked 1st followed by microarray and biosensors (Figure 3).

### 3.1. ELISA

The ELISA technique was first developed by the Swiss scientists Engvall and Perlmann in 1971 by modifying the RIA method [15]. It is a quantitative analytical method that shows antigen–antibody reactions through a colorimetric assay, where an enzyme-linked conjugate and substrate are used to identify the presence of a specific concentration of the target molecule in biological fluids. In an ELISA assay, molecules such as peptides/proteins, hormones, vitamins, and drugs are coated in the polystyrene plate, which display a very high level of specificity against their cognate antibodies or antigens. Thus, ELISA assays are considered to be a very specific assay for quantification, where target antibodies or antigens can be measured in very low concentrations with hardly any risk of interference. Synthetic peptide-based ELISA can be developed by the three ways mentioned below: (1) target antibodies are immobilized in wells of the microtiter plate by using the adsorption procedure, wherein antibodies are immobilized in polystyrene plates; (2) species-specific anti-IgG or protein G-mediated immobilization, wherein anti-IgG or protein G are first coated in the plate and then target antibodies are captured by either anti-IgG or protein G; and (3) peptide-based capture, wherein peptides are directly immobilized in wells of the microtiter plate by the adsorption procedure. The assay is performed inthe solid phase of the microtiter plates, which is generally made up of rigid polystyrene, polyvinyl, and polypropylene materials. Synthetic peptides are first adsorbed in the microplates, followed by blocking with bovine serum albumin (BSA) for uncoated sites. The common enzymes that are employed with ELISA include peroxidase and alkaline phosphatase. These enzymes are conjugated with secondary antibodies. For alkaline phosphatase, P-nitro-phenyl phosphate (PNP) are used as substrates, which produce a yellow color in positive reactions. However, for the peroxidase conjugate, 5-amino salicylic acid and orthophenylenediamine (OPD) are used as the substrates, which produce a brown color in a positive reaction. The enzyme–substrate reaction is usually completed within 30–60 min. Sodium hydroxide (NaOH), hydrochloric acid (HCl), or sulfuric acid (H_2_SO_4_) are used to stop the reaction. The results are read at 400–600 nm on a spectrophotometer, as per the conjugate used. The technique is reviewed in more detail by Aydin in 2015 [16].

### 3.2. Microarray Technology

In late 1980, microarray technology was first developed [17]. Over time, it has become a valuable research tool for scientists and hold great promise in the field of diagnostics. Peptide microarrays are high-throughput, high-content miniature devices for immunoassays. Synthetic peptides are used as a probe in microarrays, wherein peptides are adsorbed on the surface of nitrocellulose-coated glass slides and are exposed to cellular extracts or serum or other specimens for molecular recognition events. The advantage of using microarray technology is the use of a number of different unique peptide biomarkers specific to the disease in real time. All probes can be immobilized in a random manner to ensure equal accessibility to all antibodies on the peptide microarray during epitope mapping, thus avoiding concentration-dependent effects on signal intensity. The technological concept of a peptide microarray is based on the substitution of linear epitopes of the protein with short overlapping synthetic peptides. These peptides typically consist of 10–15 amino acids and capture antigen-specific antibodies from serum samples [18].

### 3.3. Biosensors

In 1956, Leland C. Clark, Jr. has developed a biosensor to detect oxygen and later he was known as the ‘father of biosensors’. His famous invention was later called by his name: the ‘Clark electrode’ [19]. Nowadays, biosensors are very common in clinical diagnosis and a number of point-of-care technologies (POCTs) have been developed for monitoring the disease diagnosis and its prognosis. In a general scenario, sensors are coupled with high-affinity biomolecules that allow selective detection of analytes. There are more than 84,000 indexed reports on the topic of ‘biosensors’ from 2005 to 2015 on ‘Web of Science’ [20]. A normal biosensor consists of five components:(1) an analyte, which can be any target molecule that needs to be detected by the biosensor; (2) a bioreceptor, which can be any molecule that specifically recognizes the analyte, such as a peptide, protein, cells, DNA, etc.; (3) a transducer, which is an element that converts one form of energy, such as bio-recognition, into another form of energy, such as optical or electrical signals; (4) an electronic circuit, which is a complex electronic circuit that performs amplification and conversion of signals to a digital form; and (5) a display, consisting of a user friendly system for interpretation of the results, such as the liquid crystal displays on computers or a direct printer that generates numbers or curves. It is a combination of hardware and software that generates the results of the biosensor in a user-friendly manner. A biosensor is a very sensitive device for measuring signal creating by biological or chemical reactions, which is proportional to the concentration of an analyte binding to its ligand. Biosensors are employed for disease monitoring, drug discovery, disease-causing micro-organisms, detection of pollutants, and presence of bio-markers indicating the disease stage in bodily fluids (blood, urine, saliva, and sweat). The technique is well reviewed by Bhalla et al. in 2016 [20].

### 3.4. Microfluidics

The field of science and technology that is associated with the control and manipulation of liquids at the microliter level is called microfluidics. Microfluidics is one of the powerful tools that is currently tying together with clinical diagnostics and generating a highly advanced version of POCT for precise and reproducibly results. It has revolutionized laboratory approaches for biological and chemical analysis from the bench-side to miniature chips. Moreover, these types of assays arecost effective and also do not require specific training to handle the device. Principally, the concept of microfluidics was associated with a framework of complexity and robustness in 1950s. The advantages of microfluidics are a reduced sample volume, scalability, laminar flow, and, hence, highly predictable fluid dynamics, a high resolution and sensitivity, and a short analysis time, leading to its low cost. The development in microfluidic technology has created a platform for genetic and proteomic analysis at the microscale level. This development is also associated with new advancement in technology along with their respective applications in pathogen detection to POCT devices, high-throughput combinatorial drug screening platforms, schemes for targeted drug delivery, advanced therapeutics, and novel biomaterials synthesis for tissue engineering.

Since the last two decades, microfluidics has started to show its impact in clinical diagnosis. The field of microfluidics is also evolving rapidly. The state of the art of microfluidic technologies is used to address the unmet challenges in diagnostics and can expand the horizons on clinical diagnostics, disease management, and patient care. Of the various microfluidic technologies that are available in the field, some are reliable and have been tested clinically. They can contribute to bridging the gap between this emerging technology and real-world applications [21].

Some advanced in vitro models, such as “organ-on-a-chip” technology, represents a new avenue in the field of scientific research and revolutionized the field of drug screening and toxicology studies [22]. Perestrelo et al. has reviewed interesting advancement in the field of microfluidic-based devices and its applications in the biomedical field, such as the body-on-a-chip concept [23].

### 3.5. Multiple Reaction Monitoring

In recent years, multiple reaction monitoring (MRM) has become more pivotal in clinical research for developing strategies for precision-based medicine or patient care. Thus, MRM is now used to evaluate proteomic/peptide biomarker verification with potential applications in medical screening. In this technique, high-quality tryptic peptides are selected and validated for quantitation of the proteins, its isoforms, and its post-translational modifications. The multiplexing of selected reaction monitoring (SRM) for targeting the number of proteins in a single run is known as MRM. It is a powerful technique based on a mass spectrometric approach for absolute and relative quantification of the proteins/peptides of interest in complex biological samples. MRM is a highly selective technique with a large capacity for multiplexing (~200 proteins per analysis per run). If the cost of transition is considered, it is rapid and cost-effective because the cost of the assay development to its deployment is low. For MRM assays, a triple quadrupole (QqQ) mass analyzer is required along with tandem quadrupole mass filters (Q1, Q3) and a collision cell. All the compartments are identical quadrupoles and may be used either to filter a specific mass-to-charge (*m*/*z*) ratio or to transmit a non-resolved ion of a specific range. Usually, the first quadrupole, Q1, is set to filter a specific precursor ion, which is passed through a collision cell and gets fragmented by the low-energy collision induced dissociation (CID) to create specific product ions. The specific product ion is detected by the Q3 analyzer for quantification. This process is referred to as the “transition process” and the technique is named “selected reaction monitoring”; the specific precursor/product ion pair is termed “transition” [24].

## 4. Peptides Application in Non-Imaging Diagnostics

In the 21st century, a number of peptide-based diagnostic systems has already been developed for commercial use or are on the verge of completion. There are a few examples of ELISAs with peptide-based diagnostic probes: C-peptide [25,26], gliadin [27], vasoactive intestinal peptide [28] diphtheria toxin (DTx) [29], *Chlamydia trachomatis* [30,31,32,33,34] human T-lymphotropic virus type I (HTLV-I) [35], human *C. pneumoniae* [31,32,33] and COVID 19 spike protein [36].

Moreover, rapid growth has been observed in peptide-based diagnostic systems mainly for diagnosis of cancer, heart disease, diabetes, Alzheimer disease, auto-immune disease, viral and bacterial infections, allergies, etc. (Figure 4). A few examples are quoted here for reference. Liu et al. has developed a novel affibody-based ELISA for detection of alpha-fetoprotein (AFP). AFP is an important biomarker associated with primary liver cancer. The peptide used in ELISA was a 58 amino acid peptide ‘Affibody’, which was derived from the Z domain of staphylococcal protein A. An affibody dimer (Z_AFP D2_)_2_ showed higher binding affinity to AFP along with high thermal stability. The detection limit of the immunoassay using (Z_AFP D2_)_2_ was 2 ng/mL [37]. Sahin et al. has selected and characterized the DE-Obs peptide HNDLFPSWYHNY by bio-panning of the phage display library on MKN-45 gastric cancer cells, which showed specific binding in MKN-45 cells [38]. Liu et al. has identified a 7-mer peptide that has the potential to be developed into a diagnostic test for residual hepatoma cells after trans-arterial chemoembolization [39]. Zhang et al. has demonstrated that the peptide sequence AADNAKTKSFPV has the potential to specifically recognize gastric cancer and discriminate neoplastic gastric mucosa from normal gastric mucosa. This can be used for early cancer detection during endoscopy [40]. Galvis-Jiménez et al. has developed an ELISA test to detect mammaglobin in blood samples from breast cancer patients vs. controls. Antibodies were generated in rabbits against four synthetic peptides of mammaglobin. All peptides showed immunogenicity and produced antibodies that were able to discriminate between the patients and controls. The results were obtained for an antiserum. B antiserum (against mammaglobin (31–39)) showed the best sensitivity (86.3%) and specificity (96%) [41].

To understanding the role of GLP-1 in diabetes and its physiology, an accurate measurement of the GLP-1 metabolite is required. In 2017, Wewer Albrechtsen et al. developed an ELISA for measurement of the primary glucagon-like peptide-1 (GLP-1) metabolite, such as GLP-1 (7-36NH_2_) and GLP-1 (9-36NH_2_). The active form of GLP-1 is (7-36NH_2_), which is rapidly degraded by the dipeptidyl peptidase 4 (DPP-4) enzyme and converts, by more than 90%, into an inactive form or to the primary metabolite (9-36NH_2_) before reaching the target via the circulation. The developed ELISA could recognize both GLP-1 (9-36) NH_2_ and nonamidated GLP-1 (9-37) [42,43,44]. The ADRB1-AB-immunogen-peptide (ESDEARRCYNDPK) impact of beta1-AAB on “myocardial recovery in patients with systolic heart failure” was published based on a peptide ELISA [45].

Increased C-peptide level is an important indicator for the diagnosis of diabetes. Lv et al. has developed an antibody sandwich ELISA for rapid detection of C-peptide in human urine of diabetic patients. Antibodies were developed in hen and rabbit by using PLL-C-peptide and BSA-C-peptide, respectively [46].

A peptide-ELISA was developed for detection of human H5N1 influenza viruses. ELISA was based on the antigenic H5 epitope (CNTKCQTP), which provides highly specific detection of antibodies to the H5N1 influenza viruses in human sera [47].

A rapid and accurate ELISA-based test was developed for HIV-1/2 antibody detection by using a peptide cocktail as an antigen. A novel peptide stretch, V3-I, covering the immunodominant epitope corresponding to the V3 hypervariable loop of gp120 antigens of selected Indian isolates, has been studied and incorporated in an antigenic cocktail of gp36, gp41, and rp24 of HIV-1/2. The peptide cocktail-based ELISA test showed 100% sensitivity and 99.3% specificity, with no cross reactivity [48]. A synthetic peptide of 11 amino acid was used to develop an ELISA for HIV-2. The peptide epitope in the ELISA was highly specific and sensitive towards anti-HIV-2 antibodies. The peptide ELISA showed 100% sensitivity with 94.9% specificity [47].

Lyme neuroborreliosis (LNB) is a disorder of the CNS caused by systemic infection of spirochetes. The diagnosis of LNB is a challenge to clinicians. Van Brugel et al. has demonstrated that the C6-peptide ELISA can be used for the diagnosis of LNB by using a patient’s CSF. Serum–CSF pairs from LNB patients (*n* = 59), Lyme non-neuroborreliosis cases (*n* = 36), and neurological controls (*n* = 74) were tested in a C6-peptide ELISA, where the sensitivity of the C6-peptide ELISA for LNB patients in CSF was 95%, and the specificity was 83% in the Lyme non-neuroborreliosis patients, 96% in the infectious controls, and 97% in the neurological controls [49].

Davis has demonstrated that an ELISA can be developed to quantify cellular proteins, such as NGF, secreted into conditioned culture media. Neurotrophin is critical to neuronal viability, and has become a popular research focus for the treatment of neurodegenerative diseases [50].

## 5. Peptide Diagnostics and SARS-CoV-2

In 2019, a new coronavirus, SARS-CoV-2, which causes acute respiratory syndrome, began to spread around the world. The disease is known as COVID-19 (coronavirus disease 2019) and has so far caused the deaths of about 4 million people worldwide and more or less serious health problems for hundreds of millions more. It is clear that the need to establish the right diagnostic and therapeutic approach is critical. The basis for a successful fight against this pandemic is not only the determination of the most effective therapy but also prevention based on reliable testing and vaccination. Thousands of scientists immediately began to address this new problem using a variety of methodological approaches. Several of these methodologies are based on the use of peptides. Examples of the use of peptides in studying the properties of SARS-CoV-2 and research into the resultant COVID-19 can be found below.

### 5.1. Viral Epitope Profiling of SARS-CoV-2

Analysis of viral epitopes is crucial for understanding the immunogenicity of the viral proteome, while it is critical for improving the diagnostics and production of a functional vaccine. Peptides frequently and specifically recognized by COVID-19 patients were identified by VirScan-based serological profiling and used to create a Luminex assay predicting SARS-CoV-2 exposure with 90% sensitivity and 95% specificity [51].

### 5.2. Peptides Used for Antibody Diagnostics

Actually, a large variety of SARS-CoV-2 antibody diagnostic assays are used, including immunoassays based on the large recombinant protein or vice-versa specific epitope peptides identified from the whole antigen [52]. Using peptide epitopes would be beneficial with respect to assay specificity, while large recombinant proteins also include many cross-reactive epitopes that would react with low specificity antibodies, leading to a lower specificity of the test.

### 5.3. Peptides Used for Identification of SARS-CoV-2-Derived T Cell Epitopes

The identification of SARS-CoV-2-derived T cell epitopes is of critical importance for diagnostic tools as well as for peptides vaccines. One way how to identify them is using CD4+ and CD8+ T cell depletion assays and FACS-based analysis of activation markers. Results obtained by using these methods suggested that generation of effective adaptive immunity against SARS-CoV-2 requires the participation of both CD4+ and CD8+ T cells. This finding is very important for the preparation of a functional vaccine, as it is clear that it is necessary to incorporate both HLA-I-restricted and HLA-II-restricted epitopes in peptide-based vaccines to obtain optimal vaccination [53].

### 5.4. Peptides/Proteins as a Markers of COVID-19

D-dimer is produced during lysis of crosslinked fibrin. Results of some studies suggest that the D-dimer levels can be used as a prognostic marker in patients with COVID-19 [54,55,56]. Interferon gamma-induced protein 10 (IP-10) is a small cytokine secreted by endothelial cells, monocytes, and fibroblasts, which attracts activated Tcells to the site of inflammation. In COVID-19 patients, IP-10 was overexpressed in the acute phase of the disease regardless of other clinical characteristics; therefore, it has been suggested as a potential new biomarker for SARS-CoV-2 infection. In the study, SARS-CoV-2 peptide pools covering viral proteins were used in order to identify the immune biomarkers of SARS-CoV-2 infection [57].

## 6. Imaging Diagnostics

The most common targeted molecular imaging techniques, such as PET and SPECT, are playing a very important and essential role in modern diagnostics because the information provided by them are very specific, accurate, and shows disease distribution. On the contrary, using non-specific contrast agents has a low targeting efficiency, which can be superseded by using specific probes. Recent technological development has revealed various methodologies for designing specific, smart, and accurate probes. Among all the strategies, utilization of peptide-based probes has been the most successful. For discovery of specific peptide-based probes, the commonly used methods are phage display and combinatorial peptide chemistry. They have strongly impacted the use of available targeting peptides in an efficient and specific manner. The discovered peptides are either a specific target for a variety of disease-related receptors or surrogate biomarkers. These targeting peptides are either radiolabeled or coupled with the appropriate imaging moieties and used in imaging diagnostic. For this reason, labeled peptides have soon become a part of imaging diagnostic systems.

## 7. PET and SPECT Imaging

In 1951, Wrenn Jr et al. demonstrated the use of a positron-emitting radioisotope for brain tumor localization [58]; after this, in the 1960s and 1970s, PET gradually grew as a research imaging modality [59]. However, the clinical utility of PET in neurology and oncology patients was demonstrated in the 1980s and 1990s [60,61].

PET is a non-invasive molecular imaging modality that uses radioactive tracers in pico- and nanomolar amounts for visualization and quantitation of biological processes in vivo. In brief, PET starts with an intravenous injection of a radioactive probe (i.e., compound labelled with positron-emitting isotopes) that circulate throughout the body and accumulate in the inflammatory lesions, which results in the emission of a positron. The positron ‘annihilates’ with an electron and generates two 511 keV γ-photons that travels in the opposite direction at 180 degrees. This property is called as ‘collinearity’. The two photons generated during the annihilation process are detected by the PET detector ring and known as ‘coincidence detection’. Coincidence detection makes PET imaging more sensitive compared to SPECT imaging, where the γ-rays emitted from the target lesions are measured directly by the detectors. A detailed description about the PET and SPECT techniques and its applications are discussed in detail by Signore et al. in2010 [62].

Interestingly, labelled peptides were introduced into clinic more than three decades ago, since then these are increasingly being used in clinics for diagnosis of different diseases, staging, and evaluation of therapy response. These radiolabeled probes are utilized in the most sensitive molecular imaging techniques, i.e., PET and SPECT.

Human cells overexpress several peptide receptors in the diseased condition, which works as molecular targets, and radiolabeled peptides bind to these targets with high affinity and specificity, holding great potential for molecular diagnostic imaging.

To understand the role of peptides in imaging diagnostics mainly for PET and SPECT, we have analyzed the published literature on PUBMED for last 5 decades (1 January 1970 to 31 December 2020). As per data published on PUBMED, we observed that PET has grown drastically in last two decades (Figure 5).

## 8. Peptides Application in Imaging Diagnostics

To understand the role of radiolabeled peptide in imagine diagnostics and get a clear picture of its usage in imaging diagnostics, we had critically analyzed the data for the last 10 years and 5 years on the PUBMED MEDLINE database. The specific keywords were either “PET” or “SPECT”, used in two separate searches with seven additional filters of the most common radiolabeled peptides and their analogues used in PET and SPECT imaging, such as somatostatin receptors, interleukin 2 receptor, prostate specific membrane antigen, αβ3 integrin receptor, gastrin-releasing peptide, chemokine receptor 4, and urokinase-type plasminogen receptor. We observed that there is no big difference between the data published in 5 years versus 10 years for the peptides used in PET and approximately a similar amount of data published in 5years and 10 years, except for αβ3 integrin and urokinase-type plasminogen receptor. It shows that most of the research was done in last 5 years (Figure 6). Similarly, the data obtained from SPECT had shown a similar pattern as mentioned above for PET; but, interestingly, no published data was obtained for Interleukin 2 receptor and urokinase-type plasminogen receptor in the search of the last 5 years (Figure 7).

### 8.1. Somatostatin Receptors (SSTRs)

In 1973, Roger Guillemin’s group first isolated somatostatin (SST) from an ovine hypothalamic extract and was characterized as tetradecapeptide [63]. SST is a regulatory and cyclic disulfide containing a peptide that is naturally present in 14 or 28 amino acids sequences. SST binds to somatostatin receptors (SSTRs) and regulates several physiological and cellular processes, which are expressed by many neuroendocrine cells, nerve cells, and inflammatory cells, such as lymphocytes, peripheral blood mononuclear cells, thymocytes, monocytes, and macrophages [64]. The function of SST is mediated by a family of G-protein-coupled receptors that includes five distinct subtypes, namely, SSTR1–SSTR5 [65].

Due to the wide range of interaction with SSTR in different inflammatory disease conditions and overexpression of SSTR in immune cells, inflammatory cells, and blood vessels, it was sensible to develop radiolabeled SST analogues with a different affinity for SSTRs for diagnostic PET and SPECT imaging in different oncological and inflammatory disease conditions, particularly in rheumatoid arthritis, Sjögren syndrome, and autoimmune thyroid diseases [66]. In brief, labelling of SST analogues were achieved by conjugation of the peptide with a bifunctional chelator, DOTA, NOTA, or DTPA, and afterwards labelling with a radionuclide, including ^99m^Tc (EDDA/HYNIC-TOC) or ^18^F or ^68^Ga, ^123^I, ^111^In, or ^64^Cu (DOTATATE) [67,68].

Several SST analogues have already been developed and assessed for clinical use; however, ^111^In-labelled pentetreotide (also known as ^111^In-octreotide or Octreoscan^TM^), a DTPA-conjugate of octreotide, is extensively studied in diagnostic imaging. ^111^In-pentetreotide has a high affinity for SSTR2 and SSTR5; it enters inside the cell by endocytosis, first taken by lysosomes and then moved to the nucleus. It is predominantly used for the assessment of neuroendocrine tumors and carcinoid tumors. Some other novel radiolabeled tracers for SSTRs also demonstrated decent affinity, including ^68^Ga-DOTA-TOC (more selective to SSTR2 and SSTR5), ^68^Ga-DOTA-TATE and ^64^Cu-DOTA-TATE (affinity to SSTR2), ^99m^Tc-EDDA/HYNIC-TOC (affinity to 2 and 5 type receptors), and ^68^Ga-DOTA-NOC (affinity to 2, 3 and 5 type receptors) [69,70]. A study was performed by Yamaga et al. to compare the detection rate of^68^Ga-DOTATATE PET/CT with ^111^In-octreotide SPECT/CT in medullary thyroid carcinoma patients (*n* = 15) with increased calcitonin levels but negative conventional imaging after thyroidectomy. This study revealed a high sensitivity and accuracy, 100% and 93%, respectively, with ^68^Ga-DOTATATE PET/CT, while ^111^In-octreotide SPECT/CT showed a lower sensitivity and accuracy, 46% and 53%, respectively. In the same study, the authors also performed conventional imaging (CI) that was comparable with PET/CT scans with a sensitivity of 100% and accuracy of 93%, although ^68^Ga-DOTATATE PET/CT demonstrated a higher detection rate compared to CI in detecting bone metastases [71].

Nevertheless, Johnbeck et al. performed a head-to-head comparison of the diagnostic accuracy of PET/CT scans of ^64^Cu-DOTATATE with ^68^Ga-DOTATOC in neuroendocrine tumor patients (*n* = 59). In this study, the authors found that 701 lesions were concordant in both PET/CT scans; however, only one of these scans detected an extra 68 lesions. The authors concluded that the patient-based sensitivity was the same for ^64^Cu-DOTATATE and ^68^Ga-DOTATOC PET/CT scans in these patients. However, ^64^Cu-DOTATATE had a comparatively better lesion detection rate, as patient follow-up discovered that the majority of the extra lesions detected by ^64^Cu-DOTATATE were true positive [72]. Nevertheless, the study also revealed thata >24 h shelf life and at least 3 h scanning window makethe ^64^Cu-DOTATATE PET/CT scan more convenient to use in the clinical setting.

### 8.2. Interleukin-2 Receptor

Interleukin-2 (IL-2) has a very high affinity for interleukin-2 receptors (IL-2R), which is hetero trimers of the α, β, and γ subunit, named CD25, CD122, and CD132, respectively. The α-subunit, i.e., CD25, contains the key binding site for IL-2, which could be present as a soluble or transmembrane receptor [62]. High levels of IL-2R are expressed by activated lymphocytes during inflammatory processes, whileIL-2R expression is lower in resting immune cells; therefore, this receptor is an appropriate biomarker for the diagnosis of active inflammation in chronic inflammatory disease patients.

IL-2 is one of the most studied research radiotracer for imaging of infiltrating T cells in different inflammatory diseases, and radiolabeled with different radionuclides, including^99m^Tc, ^123^I, ^125^I, ^35^S, and, recently,^18^F for PET imaging [62].

Signore et al. performed a study to evaluate in vivo the binding of ^99m^Tc-IL2 with infiltrating lymphocytes in thirty patients with cutaneous lesions suspected of being melanoma [73]. In this study, histology revealed 21 melanoma lesions and 9 classified as benign. The authors reported ^99m^Tc-IL2 uptake in 15 out of 21 (71%) melanomas lesions and 2 out of 9 (22%) benign cutaneous lesions. Additionally, in the ^99m^Tc-IL2 scan, the target-to-background ratio significantly correlated with the number of IL2R-positive tumor-infiltrating lymphocytes. This study demonstrated that the ^99m^Tc-IL2 scan might provide a tool for the in vivo assessment of tumor-infiltrating IL-2R-positive cells, which could be extremely beneficial for patient selection with unlabeled IL2 immunotherapy.

For PET imaging of IL-2, a novel method was published by Di Gialleonardo et al. on how to synthesize *N*-(4-Fluorobenzoyl)-interleukin-2 (FB-IL2) that specifically binds to IL-2R [74]. In addition, recently Khanapur et al. presented an improved synthesis of the same radiotracer [75]. To enable the use of FB-IL2 in clinical studies, a fully automated Good Manufacturing Practices (GMP)-compliant production process has been developed and published by Erik FJ de Vries’s group at University Medical Centre Groningen, The Netherlands [76]. In addition, the same group performed a clinical trial (ClinicalTrials.gov; identifier NCT02922283) in patients with metastatic melanoma (stage IV). In this study, the researchers found FB-IL2 to be safe and feasible for human patient study without any side effects, although serial PET imaging was not able to detect a treatment-related immune response in this patient cohort [77].

### 8.3. Prostate-Specific Membrane Antigen (PSMA)

Prostate-specific membrane antigen (PSMA), also known as glutamate carboxypeptidase II (GCPII), or folate hydrolase, is an integral cell-surface membrane glycosylated metalloenzymeoverexpressed in prostate carcinomas. It has 19 amino acids (AA) intracellular N-terminal domain, 24 AA transmembrane helix, and a 707 AA extracellular C-terminal domain bearing 2 zinc ions and 2 binding pockets [78].

Mannweiler et al. investigated paraffin-embedded sections of patients with primary prostate carcinoma and distant metastases (*n* = 51). The immunohistochemistry data revealed that 96% of the primary tumors and 84% of metastases showed expression of PSMA, in the advanced prostate cancer cohort [79]. While only one case, i.e., 1.9%, was entirely negative for PSMA in both the primary and metastatic tissue. Therefore, PSMA is a suitable biomarker for diagnosis, staging, and therapy response monitoring in prostate cancer patients.

First images of prostate carcinoma patient detected by ^68^Ga- labelled HBED-CC conjugate of the PSMA-specific pharmacophore Glu-NH-CO-NH-Lys (^68^Ga-PSMA) was published by Afshar-Oromieh et al. [80]. In this study, an^18^F-fluoroethylcholine (F-FECH) PET scan was unable to detect any lesions, while a ^68^Ga-PSMA PET scan revealed a lesion adjacent to the urinary bladder matched with tumor relapse.

Recently, Grubmüller et al. performed a study to evaluate simultaneous [^68^Ga]Ga-PSMA-11 PET/MRI for primary tumor-node-metastasis staging in prostate cancer patients (*n* = 122) prior to planned radical prostatectomy, compared with histology data. In this study, PSMA-PET/MRI correctly diagnosed prostate cancer in 119 of 122 patients (97.5%).The diagnostic accuracy for T staging was 82.5%, for T2 stage was 85%, for T3a stage was 79%, for T3b stage was 94%, and for N1 stage was 93% [81]. This study confirms the efficacy of [^68^Ga]Ga-PSMA-11 PET/MRI for an accurate staging of newly diagnosed prostate cancer patients.

Another study was performed to compare the metabolic features of high-grade glioma (HGG) and low-grade glioma (LGG) tumors using ^68^Ga-PSMA-617 and ^18^F-FDG PET scans. In this study, the patients (*n* = 30) underwent both ^68^Ga-PSMA-617 and ^18^F-FDG PET scans over two consecutive days and then surgical treatment was performed. This study revealed that the ^68^Ga-PSMA-617 PET scan is superior to the ^18^F-FDG PET scan in differentiating HGG and LGG [82].

Nevertheless, several PET and SPECT studies with radiolabeled PSMAPET radiotracers were also performed in patients with thyroid cancer [83], hepatocellular carcinoma [84], prostate adenocarcinoma [85], glioblastoma [86], myeloma [87], sinonasal glomangiopericytoma [88], Sjögren syndrome [89], and bladder cancer [90], which shows PSMA-targeted radiotracers are now playing an increasing role in the diagnostic imaging of patients.

### 8.4. αvβ3 Integrin Receptors

Integrinsare a class of 24 heterodimeric transmembrane glycoproteins made up of different 18 α-subunits and 8 β-subunits, and play a key role in cellular interactions and transduction of signals between the extracellular matrix and interior of the cell [91]. The αvβ3 integrin, also referred as the vitronectin receptor, plays a key role in tumor metastasis and angiogenesis, so diagnostic examination with αvβ3 expression offers a great prospective strategy. On the surface of vitronectin, αvβ3-binding was mediated by RGD tripeptide, i.e., Arg-Gly-Asp, which acts as the core recognition motif [92]. In diagnostic PET and SPECT imaging, αvβ3 integrin is the most extensively studied integrinthat provides crucial information about the metastatic potential of tumor, and offers an optimal in vivo biomarker for angiogenesis.

Galacto-RGD is a radiolabeledαvβ3 antagonist that helps in the monitoring of αvβ3 expression with PET imaging and the first of its class studied in human patients. Haubner et al. radiolabeled glycosylated RGD-peptide (Galacto-RGD) using 4-nitrophenyl 2-fluoropropionate as a prosthetic group with a radiochemical yield of 85% and a high radiochemical purity of >98% [93]. Afterwards the same group performed a PET imaging study in nine patients, who suffered from either malignant melanoma with distant/lymph node metastasis, or chondrosarcoma, or soft tissue sarcoma, or osseous metastasis of renal cell carcinoma, or villonodular synovitis. Researchers selected these patients based on substantial evidence that these pathologies express αvβ3 [94]. This study demonstrated a 9-fold higher radiotracer accumulation in the tumor than in the muscle, which confirms the superior properties of Galacto-RGD for molecular imaging of αvβ3 integrin receptors.

A feasibility study is recently performed by Makowski et al. using Galacto-RGD PET/CT imaging in patients with acute myocardial infarction (*n* = 12) for αvβ3 expression assessment [95]. In this study, Galacto-RGD uptake significantly correlated with infarct size (*R* = 0.73). In addition, the authors found significant inverse correlation with restricted blood flow for all myocardial segments (*R* = −0.39) and in severely hypo-perfused areas (R = −0.75).

An SPECT tracer, NC100692, was evaluated for imaging αvβ3 expression in human breast cancer patients by Bach-Gansmo et al., where the authors were able to clearly detect 19 of 22 tumors using this tracer, which was safe and well tolerated by these patients [96].

Another novel integrin-targeted PET imaging radiotracer Fluciclatide (also known as AH111585) was evaluated for αvβ3 and αvβ5 imaging in melanoma and renal tumors. The authors demonstrated that an increased ^18^F-fluciclatide uptake occurs at sites of acute myocardial infarction, in specific areas of subendocardial infarction, and hypokinesia associated with subsequent functional recovery [97]. Data from this study suggested that ^18^F-fluciclatide is a potentially useful imaging biomarker for PET imaging of myocardial αvβ3 integrin expression.

### 8.5. Other Peptides for PET and SPECT Imaging

A number of radiolabeled peptides are already established in clinics or are being evaluated in different phases of clinical trials for PET and SPECT imaging of various inflammatory diseases. Apart from the above mentioned peptides, many other peptides are also under investigation, which includes but are not limited to peptides for bombesin (BBN) receptors, gastrin-releasing peptide (GRPR), chemokine receptor 4 (CXCR4), urokinase-type plasminogen receptor (uPAR), glucagon-like peptide receptor 1 (GLPR1), and caspase-3 imaging [98,99]. 

Recently, Kraus et al. evaluated the possibility of the C-X-C motif chemokine receptor 4 (CXCR4)-directed imaging with ^68^Ga-Pentixafor PET/CT, to diagnose and quantify disease involvement in 12 myeloproliferative neoplasms patients, together with 5 non-oncologic control patients. Study data revealed that 12 out 12 patients were found positive in PET/CT, which was also confirmed by immunohistochemical staining [100]. This is the first data that shows the feasibility of CXCR4-directed imaging with ^68^Ga-Pentixafor PET/CT in a myeloproliferative neoplasm patient cohort. In the beginning, CXCR4 was recognized as a co-receptor in human immunodeficiency virus-1 (HIV-1), which attracted the researchers’ attention; afterwards, more investigation revealed overexpression of CXCR4 in 30 different cancers, including pancreatic, breast, lung, colorectal, prostate, ovarian, and skin cancers, lymphoma, and leukemia [101,102].

Zhang et al. published the first-in-human study of a ^68^Ga-labeled heterodimeric peptide BBN-RGD [103]. This novel radiotracer, ^68^Ga-BBN-RGD, targets α_v_β_3_integrin as well as GRPR. In this study, the authors investigated the diagnostic accuracy and safety of ^68^Ga-BBN-RGD PET scans in healthy volunteers (*n* = 5) and prostate cancer patients (*n* = 13) and comparedit with ^68^Ga-labelled BBN. This study did not show any obvious side effect of ^68^Ga-BBN-RGD administration in any healthy volunteer and/or patient. In patient scans, ^68^Ga-BBN-RGD PET/CT diagnosed 3 of 4 primary tumors, while only 2 of 4 primary tumors were diagnosed with ^68^Ga-BBN PET/CT. Interestingly, the authors found that ^68^Ga-BBN was not able to detect lesions that were GPRR -ve and α_v_β_3_ integrin +ve; however, ^68^Ga-BBN-RGD was able to detect these lesions. Therefore, this novel approach for dual α_v_β_3_ integrin and GRPR, targeting PET radiotracers, demonstrated a great potential in diagnosis and staging of primary prostate cancers as well as metastases lesions. Recently, another novel dual-targeting ^68^Ga-NODAGA-LacN-E[c(RGDfK)]_2_Glycopeptide has also been developed for PET imaging of cancer patients, which can diagnose integrin α_v_β_3_ and galectin-3 expression in tumor and tumor endothelial cells [104]. As evident from the details above, enough novel radiolabeled peptides are now available in the clinical setting and many more peptides are currently in preclinical investigation and indifferent clinical trial stages; therefore, there is no doubt that it will have enormous clinical impact in diagnostics in the coming years.

## 9. Challenges in Peptide-Based Diagnostics

The major challenges in peptide-based diagnostics are the synthesis and purification of peptides. Fmoc/tBu strategies are widely used for solid-phase peptide synthesis (SPPS). The first amino acid is coupled to the resin. The first step is to deprotect the amine and then coupled with the free acid of the second amino acid. This cycle repeats until the desired sequences have been synthesized. SPPS cycles may also include capping steps, which block the ends of the unreacted amino acids from reacting. At the end of the synthesis, the crude peptide is cleaved from the solid support, while simultaneously removing all protecting groups using a strong acid reagent, such as trifluoroacetic acid or a nucleophile. The crude peptide can be precipitated from a non-polar solvent such as diethyl ether in order to remove soluble organic by-products. The crude peptide can be purified using reversed-phase HPLC [105,106].

To purify the longer peptides is very challenging, because the impurities of the by-products have somewhat similar sequences and shows the same retention time in HPLC purification. Kent and co-workers have proposed that some peptide sequences for intra-molecular or inter-molecular non-covalent interactions, which ultimately cause insoluble peptide aggregates, are not easy to solubilize for purification [107]. Moreover, synthesis of longer peptide sequences >50 amino acids has always been a challenging task for chemists, even when using well-advanced and automated peptide synthesis systems. Difficult peptide sequences are hydrophobic in nature and contain a large number of β-branched amino acids along with leucine, valine, phenylalanine, or isoleucine, etc. Peptide sequence with glycine may induce β-sheet packing. These type of peptide sequences can form β-sheet or α-helical structures within the molecule and therefore they have high aggregation potential and low solubility in aqueous or organic solvents. This results in generally difficult handling, synthesis, and purification [106].

Beside this, there are some challenges that depend upon the technique used for diagnosis. For example, non-imaging techniques, such as ELISA, microarray, and biosensor, are (1) a labor-intensive process—to prepare antibodies through a sophisticated cell culture technique; (2) high purity primary antibodies or synthetic peptides are required to set-up the experiments; (3) due to insufficient blocking of the surface onthe microtiter plate there is high possibility of getting false-positive or false-negative results; (4) antibodies instability during transportation and storage may cause false-negative results; (5) unavailability of specific antibodies or difficult peptide synthesis may cause a problem in setting up the process; and (6) it is a time-consuming process, requiring at least 24 h to complete [108].

The main limitation to set up the analysis is depended upon substances such as antigen-specific antibodies, a peptide epitope for specific antibodies, and its purity. Impure substances may cause either false-positive results or a low signal-to-noise ratio. However, imaging diagnostics are also having quite similar challenges but is seen to be more specific than non-imaging diagnostic because only labeled peptide probes are used.

As per the guideline for method validation ICH Q2 (R1), specificity is defined as follows: “Specificity is the ability to assess unequivocally the analyte in the presence of components, which may be expected to be present” (European Medicines Agency, ICH Topic Q 2 (R1), accessed on 12 August 2021) [109].

The specificity of the method is based on detection of a single or specific analyte, showing no cross reactivity to other molecules. This type of approach can be achieved by using unique peptide probes discovered or designed for specific target recognitions. These are used as a ligand in non-imaging diagnostics and as a tracer in imaging diagnostic. However, in some cases high similarities in analytes are observed, where a selective approach is considered to be the best solution that can be achieved.

For an adequate target-to-background ratio, the probe should be highly specific towards its receptor and should show a high binding affinity. Moreover, it should be functionally stable in physiological conditions and be cleared quickly from non-targeted sites in order to provide high-quality results. Additionally, it is also necessary to check the toxicity and immunogenicity of the probe for clinical translation [110].

In conclusion, peptide-based diagnostics is an interdisciplinary approach, for which scientists are performing basic research to discover unique peptides for targeting specific receptors reflecting a disease state, organic chemists are developing and characterizing the peptide-based probes, and biophysicists are improving image quality. Finally, clinicians are reviewing the outcome and importance of the diagnostics methods developed.

## Figures and Tables

**Figure 1 ijms-22-08828-f001:**
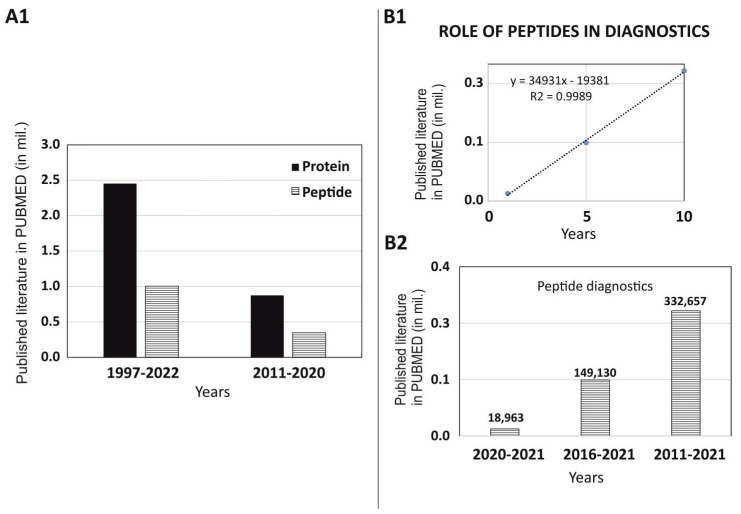
Role of peptides in diagnostics based on scientific research published on PUBMED: (**A1**) comparison of the published data of diagnostics using protein versus peptides-(**B1**,**B2**) exploring the role of peptides in diagnostics (1 year, 5 years, and 10 years).

**Figure 2 ijms-22-08828-f002:**
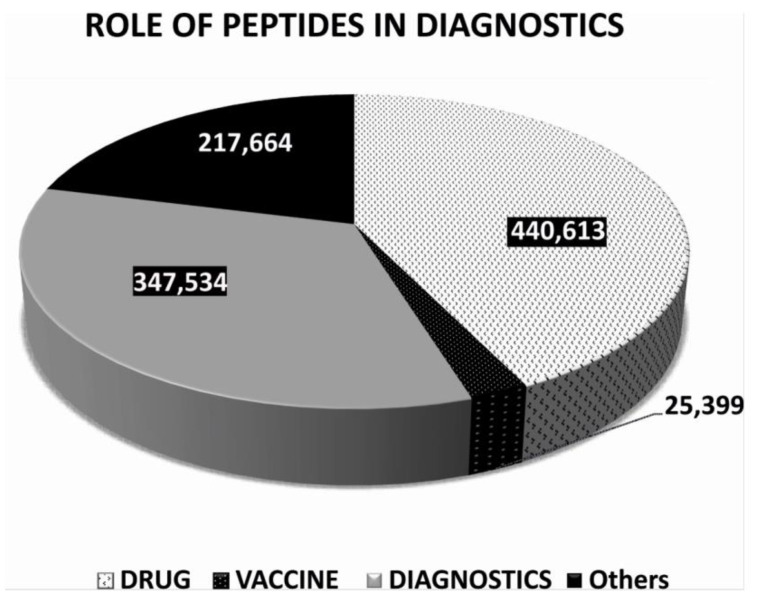
Based on scientific research published on PUBMED in the last decade (1 January 2011 to 31 December 2020).

**Figure 3 ijms-22-08828-f003:**
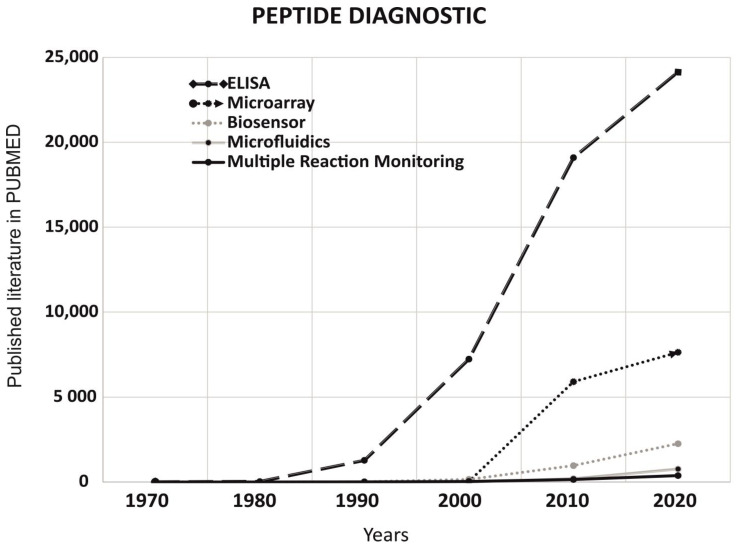
Understanding the role of peptides in diagnostics through the published literature on PUBMED in the last 5 decades (1 January 1970 to 31 December 2020). Comparison of the data published with the keywords “Peptide diagnostic” along with additional filters for ELISA, Microarray, Biosensors, Microfluidics, and Multiple Reaction monitoring. We observed ELISA ranked 1st followed by Microarray and Biosensors.

**Figure 4 ijms-22-08828-f004:**
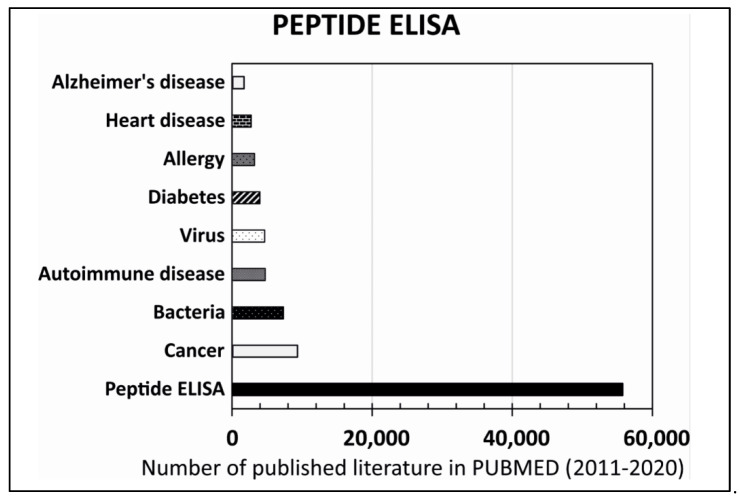
Understanding the role of peptides in ELISA in diagnostics through the published literature on PUBMED in the last decade (1 January 2011 to 31 December 2020). Comparison of the data published with the keyword peptide ELISA along with additional filters for Alzheimer’s disease, Heart disease, Allergy, Diabetes, Virus, Autoimmune Disease, Bacteria, and Cancer, respectively. We observed Cancer was ranked 1st for peptide ELISA.

**Figure 5 ijms-22-08828-f005:**
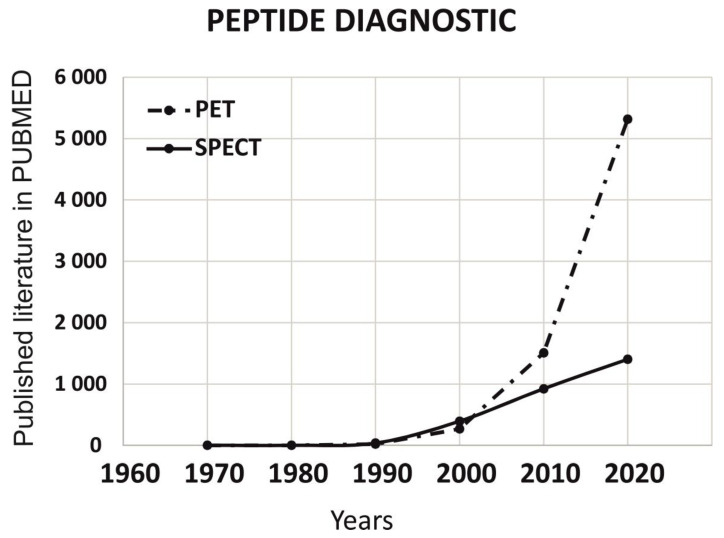
Understanding the role of peptides in imaging diagnostics (PET and SPECT) through the published literature on PUBMED in the last five decades (1 January1970 to 31 December 2020). Comparison of the data published with the keywords peptide diagnostic and sub-keywords (PET and SPECT). PET has grown drastically in last decade.

**Figure 6 ijms-22-08828-f006:**
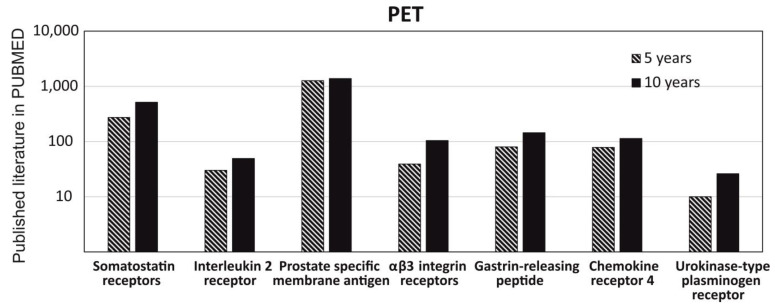
Understanding the role of radiolabeled peptide in PET through the published literature on PUBMED in the last 5 years and 10 years. Comparison of the data published with different keywords, such as somatostatin receptors, interleukin 2 receptor, prostate specific membrane antigen, αβ3 integrin receptor, gastrin-releasing peptide, chemokine receptor 4, and urokinase-type plasminogen receptor.

**Figure 7 ijms-22-08828-f007:**
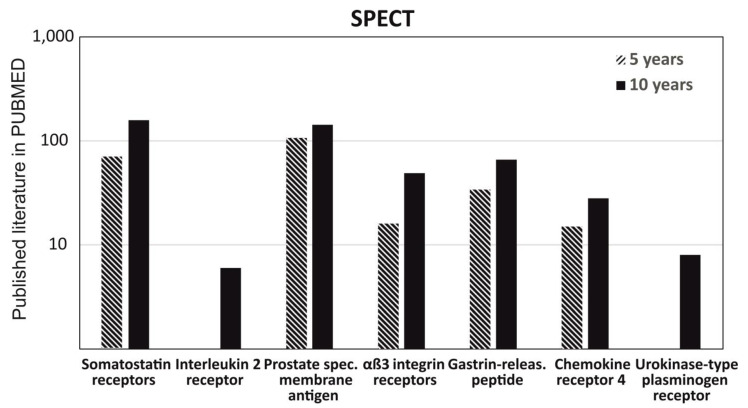
Understanding the role of radiolabeled peptide in SPECT through the published literature on PUBMED in the last 5 years and 10 years. Comparison of the data published with different keywords, such as somatostatin receptors, interleukin 2 receptor, prostate specific membrane antigen, αβ3 integrin receptor, gastrin-releasing peptide, chemokine receptor 4, and urokinase-type plasminogen receptor.

## Data Availability

Not applicable.

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
