# Peer review of "Role of Peptides in Diagnostics"

_ijms, 2021, doi:10.3390/ijms22168828_

Round 1

Reviewer 1 Report

This Review is brief and describes the most important non-imaging and imaging methods commonly used in medicine based on taking advantage with peptides or their complexes. The most important discoveries of the last decade are described and summarized. Nevertheless, a few additions would be indicated that could increase the value of the article. I have listed them in the points below:

  1. A large part of the work is devoted to specific preparations for diagnosis, drugs or peptide mediated connected with disease entities, such as SAR-Cov-2 or cancer. In my opinion it will be worthy to add some information (few sentences) about it in Abstract. Mention in abstract about applicalible peptides connected with imaging diagnosis as SSTR, interleukin-2 and αvβ3 integrin receptors, in a general way.  

  1. It's worth adding to comment on the described methods (ELISA, MT and biosensors) regarding their limitations and disadvantages.

  1. In review manuscript there is a lot of punctuation errors, that hinder reading comfort. The require correction before publication, some of them are listed below according by row no:

82 – type and size of name font should be should be like regular style;  

15, 22, 82, 92, 113, 114, 162, 164, 186, 196, 207 (doble), 208, 269, 295, 327, 364, 389, 393, 395, 396, 401, 402, 426, 438, 495, 525, 556, 558, 572, 580, 584, 590, 611, 614, 604 - space missing;

107, 108, 112, 113 – bold type and big size of font are not necessary;  

284, 320, 334, 399, 448, 463, 500 – coma before ‘and’, ‘but’, ‘or’ and ‘is’ is not necessary;  

430, 571 – missing ‘dot’,

563 – ‘α- subunits’ should be without space ‘α-subunits’,

571 – ‘in-vivo’  should be italic font,

In many cases before reference (citation) the space is missing too.

Moreover the correct name of ‘5 amino salicylic acid’ should be change to ‘5-aminosalicylic acid’ at line 157.

  1. Also  a few corrections in figures have to be made and they are as follows:  

In figure 1 A1 the year on y-axis rather should be ‘1997’ not ‘1797’;

In figure 6 is not the word ‘literature’ is illegible and there is missing space in ‘αβ3 integrin’;

In figure 7  is missing space in ‘αβ3 integrin’.

Reviewer 2 Report

Peptide based diagnostic become of the popular diagnostic tools in biomedical sciences. In recent years many advantages has been discussed in detail on many series of reviews on different platform.

 In this present review Author discussed the detail methods for role of peptide in imaging and non-imaging diagnostics. The figure 1 discussing the usage of peptides in many clinical and diagnostic approaches in very intriguing and discussed in detail in other part of the paper.

But, the challenges in peptide synthesis, lifetime and solubility is uncovered in this article, in addition it would be great is target specificity and selectivity is discussed in detail.
